# The Native Ant *Lasius niger* Can Limit the Access to Resources of the Invasive Argentine Ant

**DOI:** 10.3390/ani10122451

**Published:** 2020-12-21

**Authors:** Marion Cordonnier, Olivier Blight, Elena Angulo, Franck Courchamp

**Affiliations:** 1Ecologie Systématique Evolution, CNRS, AgroParisTech, Paris-Saclay University, 91405 Orsay, France; elena.angulo@universite-paris-saclay.fr (E.A.); franck.courchamp@universite-paris-saclay.fr (F.C.); 2Aix Marseille Univ, CNRS, IRD, IMBE, Avignon University, 84000 Avignon, France; olivier.blight@imbe.fr

**Keywords:** aggression, Argentine ant, dominant species, interference competition, invasive species, resource exploitation

## Abstract

**Simple Summary:**

Invasive ants are often highly dominant competitors, having strong impacts on native species. Such invaders often exploit resources better than native species, finding them first or collecting them faster. They are also often more efficient when interfering with other species, suffering fewer losses or preventing access to resources. We assessed the competitive behavior of the invasive Argentine ant when facing another invasive species or a native dominant species. The exploratory behavior of the Argentine ant was strongly inhibited by the native dominant species. The Argentine ant brought very few prey resources to its nest and killed few opponents. Conversely, the other invasive species had low impact on the Argentine ant. Contrary to expectations, the invasive species lacked the ability to hinder resource exploitation by the Argentine ant, whereas the native dominant species did. These results suggest that a native species could impact invasive populations of the Argentine ant by interference competition, perhaps better so than some invasive species. In the northern half of Europe, it could prevent further expansion of this highly invasive species.

**Abstract:**

Within ant communities, the biotic resistance of native species against invasive ones is expected to be rare, because invasive species are often highly dominant competitors. The invasive Argentine ant (*Linepithema humile* (Mayr)) often demonstrated numerical dominance against its opponents, increased aggressiveness, and ability to quickly recruit to food. The present study aimed to assess the behavioral mechanisms involved in the interspecific competition between *L. humile*, facing either an invasive species (*Lasius neglectus* Van Loon, Boomsma and Andrásfalvy) or a native dominant species (*Lasius niger* (Linnaeus)). The resource exploitation by the Argentine ant was investigated during one-hour competitive interactions using 10 dead *Drosophila* flies as prey. When facing *La. niger*, *L. humile* exploratory behavior was strongly inhibited, it brought very few prey resources, and killed few opponents. Conversely, *La. neglectus* had a low impact on *L. humile*. Contrarily to expectations, the invasive *La. neglectus* lacked the ability to hinder *L. humile* resource exploitation, whereas the native *La. niger* did. These results suggest that *La. niger* could impact invasive populations of *L. humile* by interference competition, perhaps better so than some invasive species. While *L. humile* has become invasive in Southern Europe, the invasion process could be slowed down in the northern latitudes by such native dominant species.

## 1. Introduction

Competition between species occurs in two ways. Exploitation competition involves the ability of species to find and exploit rapidly a resource before others, thereby making it unavailable to competitors. Interference competition involves the ability of species to prevent resource use by others (or to expulse them from the resource), either directly by aggression or indirectly by maintaining a territory [1,2]. Within ant communities, species co-occurrence could be partly explained by the fact that each species is assumed to excel in either competition by interference or competition by exploitation. Species depending on similar resources can coexist by means of a trade-off between the species’ ability to dominate resources and to discover them. This discovery–dominance trade-off occurs when species’ ability to excel at interference competition results in specialized morphological, behavioral, and physiological characteristics that reduce its ability to discover resources in the first place [3]. For instance, through competition by interference, most ant species affect abundance, spatial distribution, and behavior of other species through aggressive techniques that range from the use of chemical repellents to the establishment of territories [4,5]. Ant species can therefore be classified as dominant, subordinate, and submissive species [6]. Invasive ants especially are often highly aggressive, dominant competitors that displace many native species, through both interference and exploitative competition [7,8,9]. The dominance–discovery trade-off is indeed broken by these invasive species [5,10,11]. Finally, because the species previously established is more familiarized with the nesting and foraging site (hereinafter called “resident effect”), such species could have an advantage during the competitive interactions [12], for instance due to local numerical dominance.

During the invasion process, the resistance of local communities mainly depends on the presence of dominant ants [13,14] rather than species diversity [15]. The competition between invasive and native ant species has been substantially studied for a long time [7,16]. The interactions between the invasive Argentine ant (*Linepithema humile* (Mayr)) and native species have demonstrated that the competitive ability of *L. humile* stems from numerical dominance, aggressiveness, superior interference and exploitation competition, and the ability to quickly recruit to food [7,16]. For instance, Carpintero and Reyes-López [9] conducted a bait experiment where *L. humile* aggressively displaced large numbers of native ant species from the bait, whereas native species did not. In the study of Buczkowski and Bennett [12], Argentine ants aggressively outcompeted the native ant *Tapinoma sessile* (Say) from the baits through efficient interference competition and monopolized bait resources (exploitation competition). Studies considering the competition of invasive species facing dominant native species, and especially facing other invasive species, are less frequent. However, although invasions by the Argentine ant lead to an almost systematic exclusion of native ants and decreased ant community richness, some dominant native ants can resist (e.g., *Iridomyrmex rufoniger* (Lowne) [17]; *Tapinoma* group *nigerrimum* [18]). Similarly, some invasive species can also resist, as shown for *Solenopsis invicta* (Buren), where the competitive outcomes depended on the number of workers in each colony [19].

The present study therefore aimed to assess the behavioral mechanisms involved in the interspecific interference competition between invasive species or between an invasive and a native dominant species through laboratory experiments. More precisely, the main objective was to study how a competitor modulates resource exploitation using one the most successful invaders, the Argentine ant (*L. humile*) accounting for the resident effect of the established species. We studied interactions between *L. humile* workers facing workers, either of the invasive *Lasius neglectus* Van Loon, Boomsma and Andrásfalvy or of the native dominant *Lasius niger* (Linnaeus) species. Although most studies of the competitive ability of *L. humile* measured competition between individual ants or very small groups of workers, we conducted competition tests with groups of one hundred individuals to capture the group effect [18]. We also studied the effects of either a resident or colonizer status (i.e., familiarized or not with the foraging arena). *Lasius neglectus* is, as *L. humile*, one of the 19 species listed as highly invasive by the IUCN invasive species specialist group [IUCN SSC Invasive Species Specialist Group, 2019]. It is a more recent invader, described as a new species in 1990 [20], and shown to have a behavioral superiority over *L. humile* [21]. Both species are widely distributed across Europe and interact with invaded native communities, which often include the native black garden ant *La. niger*, one of the most common and dominant species in Europe. *Lasius niger* is characterized by its opportunism, its aggressiveness, and its ability to use mass recruitment [22,23], giving it a strong competitor’s potential. We measured 33 behavioral descriptors of the interactions covering the space occupation, the ability to exploit resources and the aggressiveness of *L. humile*, as well as the strength of the competition by the other species. We used controlled laboratory experiments where the behavioral characteristics that discriminate against the competitive ability of the three species can be assessed in the absence or presence of the competitor. As an invasive species, we expected to find evidence that *La. neglectus* has greater competitive abilities than *La. niger*, and therefore may hinder *L. humile* resource exploitation. The species established before the other, and therefore more familiarized with the nesting and foraging site, was also expected to have an advantage in the competitive interactions.

## 2. Materials and Methods

### 2.1. Studied Species

Three distinct species have been studied: the Argentine ant *L. humile*, the invasive garden ant *La. neglectus* and the black garden ant *La. niger*. *Linepithema humile* is currently a widespread and abundant invasive species, forming supercolonies (arising from the low levels of intraspecific aggression among colonies), with polydomic, highly polygynous nests and totally sterile workers [7]. *Lasius neglectus* also forms polygynous colonies with the presence of several functional queens within the nest [24]. This invasive species has ecological impacts on the biodiversity of Formicidae and other invertebrates (e.g., reducing the spatial and temporal foraging of native ants; [25,26]). The range of *La. neglectus* has increased rapidly and steadily in non-native locations over the last 30 years [27]. These two invasive species are present across almost all Europe and their distribution partly overlaps, especially in the easternmost part of Europe (Appendix A). *Lasius niger* is a widespread monogynous species, inhabiting all of Europe and parts of Asia and North America and colonizing a broad diversity of environments, but particularly abundant in arable land, as well as in cities, parks and gardens [23,28]. This species is aggressive towards other ants, including conspecifics [29].

### 2.2. Biological Material

The biological material used for the experiments came from colonies sampled between 23 April 2019 and 5 July 2019, from different sites: two super-colonies of *La. neglectus* (LnA—45.8248, 5.1004, Balan, France, collected on 19 June and 4 July 2019 and LnB—45.785258, 4.872009, Villeurbanne, France, collected on 21 June and 4 July 2019), two super-colonies of *L. humile* (LhA—36.992291, −6.451077, Huelva, Spain, collected on 23 April 2019, LhA’—43.209564, 5.629815, La Ciotat, France, collected on 15 May 2019 and LhB—43.402946, 6.730125, Fréjus, France, collected on 16 May 2019) and two colonies of *La. niger* from Villeurbanne, France (LniA—45.779328, 4.866604, collected 20 June and 5 July 2019 and LniB—45.781247, 4.867727, collected 21 June and 5 July 2019). In *L. humile*, the letter A indicates that it corresponds to the main supercolony and the letter B indicates that it corresponds to the Corsican supercolony [30,31]; in *La. neglectus*, the workers from colonies A and B demonstrate strong aggressiveness between each other, suggesting different supercolonies [32]. Each collected colony corresponds to three or more pooled nests on the same site. The colonies collected comprised several thousand workers, brood in substantial quantities, and at least five queens for nests belonging to polygynous species (only workers have been collected in *La. niger* monogynous colonies, preventing the colonies’ destruction).

Colony fragments were kept in the laboratory at 25 ± 3 °C with a mean hygrometry at 47%, and maintained in their original nesting substrate in boxes of 370 × 255 × H160 mm^3^. Colonies were supplied with a water-honey dilution (50%), proteinate food (insects such as mealworms, fruit flies or crickets) and water ad libitum. The interaction experiment was conducted in the two months after the installation and acclimation of the colonies.

### 2.3. Interaction Experiments

Interaction experiments were conducted between 20 June and 15 August 2019. The experimental set-up was as follows (Appendix A): two small peripheral plastic boxes (79 × 79 × H34 mm^3^) were connected by small plastic tubes (diameter 0.8 cm, length 2 cm) to a central arena (165 × 100 × H85 mm^3^). Each box was closed with stainless steel mesh to prevent leakage. These boxes were characterized by a resting site opposite the arena, with a source of humidity (small tube with wet cotton), with a 15 × 15 mm^2^ area covered with a red filter (favorable brightness). For each interaction, one hundred (±2) “naive” workers (i.e., never tested before) were taken from the donor colony, as closely as possible from the food resources to select preferentially foraging workers, and placed in a peripheral box with five brood elements. Water was supplied ad libitum. The taken ants were starved for 72 h prior to each trial [18]. During this period, one side of the device was open, allowing exploration of the central arena by one of the groups (resident), whereas the other did not have access to the arena (colonizer). The entrance of the resident was re-closed one hour prior to the prey being added.

Prior to the experiment, 10 dead *Drosophila melanogaster* (Meigen) individuals (adequate prey in field experiments with *L. humile* and native species; [33]) were placed in the center of the arena, equidistant from the two groups. At the start of the trial, the two entrances to the arena were simultaneously opened, and recruitment and interactions were filmed for one hour with two consumer-electronics RGB cameras BRIO 4K Ultra HD (Logitech, Lausanne, Switzerland). On each movie recorded, the species, the source colony, the date, and the conditions (temperature, hygrometry, status) were indicated on the video. All combinations of interactions with *L. humile* facing (i) no opponent, (ii) *La. neglectus* opponent and (iii) *La. niger* opponent were tested for both colonies of each species and successively for resident and colonizer statuses. Each individual test was replicated three times, resulting in 60 one-hour interactions recorded, covering a wide panel of situations encountered by a new colonizer, e.g., from no competitor (n = 12 videos), to an invasive or native competitor (n = 48 videos).

### 2.4. Video Analyses

Each video was analyzed by one of three observers familiar with the species and the ant’s behaviors. To ensure the consistency of results, ten percent of the videos were monitored by two observers simultaneously and ten percent of the videos were examined by two observers separately. All the videos have been analyzed twice (one time per interacting species). For each analysis, several global metrics were measured separately for each species: the time before the first worker leaves the peripheral box, the time between the entrance in the arena and the discovery of the resource and the time before the first interspecific interaction. We systematically recorded the temperature and hygrometry of the testing environment. Kinetic metrics were measured every five minutes for each species: the number of ongoing fights in the arena, the number of dead workers, the number of preys brought into the nest, moved in the arena or still available on the bait, the number of workers in the whole arena and the number of workers on the bait.

### 2.5. Statistical Analyses

All statistics were carried out using R v. 3.3 (RC Team, Vienna, Austria) software. Based on the kinetic and global variables resulting from the video analyses, 33 ethological descriptors of the interactions have been summarized in four categories, reflecting various aspects of the interactions: the space occupation by *L. humile* (5 descriptors), *L. humile*’s ability to exploit resources (8 descriptors), the aggressiveness of *L. humile* (8 descriptors) and the strength of the competition by the other species (12 descriptors; for interactions with opponents only; Appendix B, Table A1). We reduced these variables following three consecutive procedures: (i) remove correlated variables, (ii) run two principal component analyses (PCAs) for the different sets of descriptors, and (iii) run linear discriminant analyses (LDAs), based on the opponent species (coupled in previous PCAs), as explained below.

First, for each category of descriptors, we ran a collinearity analysis, eliminating the descriptors having a Spearman correlation value >0.7 with others, to establish a set of uncorrelated variables [34]. In each pair of correlated variables, the variable with the highest absolute correlation (i.e., the maximum average correlation with other variables) was identified using the find correlation function of the package caret [35] and removed (in italics in Appendix B, Table A1).

Second, the 22 remaining descriptors from the four categories (Appendix B, Table A1) have been used to perform two PCAs. The first PCA (Appendix B, Figure A1) was performed on 60 trials, including all variables regarding space occupation by *L. humile* and its ability to exploit resources, to reveal contrasts between tests with *L. humile* (i) alone, (ii) facing *La. neglectus* and (iii) facing *La. niger*. The second PCA (Appendix B, Figure A2) was performed on 48 trials, including all variables regarding the aggressiveness of *L. humile* and strength of the competition by the other species, to reveal contrasts between tests with *L. humile* facing (i) *La. neglectus* and (ii) *La. niger*.

Third, to identify variables varying according to the Argentine ant opponent (*La. neglectus*, *La. niger* or no opponent), we performed two discriminant analyses (LDAs) based on the two previous PCAs (package ade4; [36]). The significance of the eigenvalues was evaluated using a non-parametric version of Pillai’s test. Variables having a high proxy of the contribution to the discriminant function, i.e., whose cosines between the variables and the first axe of the linear discriminant analysis were >0.5 (as an absolute value) were considered the best drivers of the differences between species. The discriminant analyses suggested that eight variables were mainly responsible for the differences between the situations with different status and opponent species: the mean number of *L. humile* workers simultaneously present in the whole arena (mean_arena_Lh, M1), the number of preys brought by *L. humile* (n_totprey_Lh, M2), the standard deviation of the numbers of *L. humile* workers on the bait over time (ETbait_Lh, M3), the mean number of *L. humile* workers simultaneously present on the bait (mean_bait_Lh, M4), the number of dead *L. humile* individuals (n_deadtot_Lh, M5), the number of dead opponent individuals (n_deadtot_opp, M6), the mean number of simultaneous fights during the contest (mean_fights, M7) and the time when the maximal number of simultaneous fights occurs (t_maxfights, M8) (Appendix B, Table A2).

After reducing the 33 variables to eight, using the described three-step procedures, these eight variables were used to investigate the importance of (i) the opponent (*La. neglectus*, *La. niger* or no opponent), (ii) the status (resident vs. colonizer) and their interaction. In order to do that, we performed eight mixed models using the package glmmTMB [37]. The identity of the 20 combinations of source colonies and status was introduced in the model as a random effect. M1, M3, M7 and M8 were fitted with linear models, M2 and M4 were fitted with linear models performed following a square root transformation of the dependent variables, M5 and M6 were fitted with generalized linear models using Poisson family. The significance of each explanatory term was tested using Chi-squared tests. For every significant variable, post hoc, pairwise contrasts among treatments were performed by calculating the least square means (package lsmeans; [38]), and t-tests were adjusted to the number of tests using Tukey corrections. The raw data and scripts are available on the Zenodo repository (https://doi.org/10.5281/zenodo.4327129).

## 3. Results

Results of the PCA and LDA indicated likely differences in *L. humile* space occupation and ability to exploit resources according to the opponent species, but not regarding the colonizer/resident status, with more individuals in the arena and on the bait when alone in the arena than when in the presence of *La. niger*. The response seemed to be intermediate in the presence of *La. neglectus* (Appendix B, Figure A1, Table A2). The aggressive and competitive behaviors of *L. humile* seemed to differ against *La. neglectus* or *La. Niger*, and also differed according to status when facing *La. niger* (Appendix B, Figure A2, Table A2).

The models confirmed that six of these eight discriminant variables significantly varied among the opponent species (see below), whereas only one has been detected as significantly impacted by the status of *L. humile* (resident/colonizer) and its interaction with the opponent species (mean_arena_Lh, the mean number of *L. humile* workers simultaneously present in the whole arena) (Table 1).

When facing *La. niger*, *L. humile* were fewer in the arena and the presence of resident *La. niger* seemed to strongly inhibit the foraging activity of colonizer *L. humile* (M1; Table 1, Figure 1). Facing *La. niger*, *L. humile* brought fewer preys than when facing *La. neglectus* or alone (M2; Table 1, Figure 2a), and was present on the bait with fewer workers (M3-4; Table 1, Figure 2b-c). *Linepithema humile* fought less against *La. niger* than against *La. neglectus* (M7; Table 1, Figure 3a) and thus killed fewer opponents of *La. niger* than of *La. neglectus* (M6; Table 1, Figure 3b). The number of *L. humile* in the arena was the same with and without *La. neglectus* (M1; Table 1, Figure 1). Despite a lower number of *L. humile* on the bait when facing *La. neglectus* than when alone (M4; Table 1, Figure 2c), the occupation of the bait by *L. humile* did not differ over time (M3; Table 1, Figure 2b), as well as the total number of preys brought by *L. humile* (M2; Table 1, Figure 2a).

## 4. Discussion

Among the behavioral descriptors of the interactions between the competing ant species, eight significantly discriminated against the opponent species (invasive or native opponents) or status of the opponent (resident or colonizer). When facing *La. niger*, the exploratory behavior of *L. humile* workers was inhibited, especially when *L. humile* was colonizer. Workers of *L. humile* brought very few resources, were almost absent from the resources, engaged in fewer fights and killed fewer opponents when facing *La. niger*. In contrast, *La. neglectus* was not so impactful, not modifying the number of *L. humile* in the arena or the quantity of resources brought by *L. humile*, although *L. humile* engaged in more fights with *La. neglectus* and killed more opponents. Contrary to what we expected, *La. niger* showed greater competitive ability to hinder *L. humile* resource exploitation than *La. neglectus* (Figure 4).

Our main outcomes suggested that the native, and not the invasive *Lasius* species, may reduce the propensity of the Argentine ant to collect resources through interference competition. This finding is in contrast with the general established knowledge that aggressiveness and fight strategies of *L. humile* may explain its superiority over native ant species [8,12]. For instance, Carpintero and Reyes-López [9] showed that the Argentine ant is a competitively dominant species, because of its aggressive behavior and relative abundance, compared to native species such as *Cataglyphis floricola* Tinaut, *Camponotus pilicornis* (Roger) or *Pheidole pallidula* (Nylander) or *Aphaenogaster senilis* Mayr. When confronted with the Argentine ant, eight native species tended to retreat more frequently than Argentine ants (which also had initiated most of the encounters), which could help them to displace native species [16]. However, some cases in the literature have also found that specific local ants can offer strong resistance and delay or prevent the spread of Argentine ants, especially when encountering ecologically dominant or functionally similar native species (e.g., *T. group nigerrimum* [18], *I. rufoniger* [17], *La. grandis* Forel [14] or *Prenolepis imparis* (Say) [39]).

When facing *La. niger*, the risk perceived by *L. humile* appeared to decrease strongly its propensity to explore the arena, both for fighting and foraging, especially when exploring novel areas already colonized by *La. niger*. It has been suggested that chemicals laid in the environment could be used by *L. humile* as cues for the presence of local species, as they play an important role in the interactions [39]. A behavioral response to allocolonial or allospecific footprint cues could prevent encounters of potential competitors and thus be beneficial by reducing costs from competition [40]. Subordinate ant species avoided cuticular hydrocarbons of dominant species [41]. *Lasius niger* strongly differentiated between different cue types and avoided cues of allospecifics and allocolonial conspecifics [40]. In the present study, *L. humile* could thus have used chemical signals as well as direct allospecific interactions to evaluate the risk, and limited more its exploration activity when resident *La. niger* were present than when *La. neglectus* was present. By decreasing the ability of *L. humile* to collect resources, *La. niger* could also limit its expansion. This result is even more important as this dominant species is widely established in many environments, and the Argentine ant could therefore be challenged by habitats already occupied by the native ant *La. niger* when spreading north along Europe.

Conversely, some invasive species could be ineffective when facing *L. humile*, such as *La. neglectus* in the present study. For instance, *T. magnum* Mayr has shown considerable potential as an invasive species in Northern Europe, where it thrives in many urban areas and is considered a pest [42]. A recent study showed that *T. magnum* was systematically excluded by *L. humile* from resources and underwent a visible reduction in activity [43]. The foraging activity of *L. humile* was nevertheless reduced in the presence of *T. magnum* due to an increased worker commitment, both in the arena fights and directly in the colony of the competitor. In the present study, the native species *La. niger* was not excluded from the arena by *L. humile*, and its presence also resulted in a reduced foraging activity of *L. humile* due to an increased worker commitment in the arena fights. Moreover, Leonetti et al. [43] suggested that the nest of the opponent could have been perceived as a new potential threat or resource and, therefore, a more valid target for recruitment than the trophic resource. However, this result could differ for other food contents, such as carbohydrate food that could be more attractive than protein sources [44]. For instance, *La. niger* showed clear avoiding behavior when encountering the dominant species *Formica fuscocinerea* (Forel) at a carbohydrate-rich food source [45].

Our results showed that *La. niger* suffered very few losses during interactions with colonizer Argentine ants (mortality <5%), especially compared to *La. neglectus* (mortality ~10%). This finding contrasts with Bertelsmeier et al. [21], where dyadic interactions between *La. neglectus* and *L. humile* led to 90% mortality for *L. humile* and 35% for *La. neglectus*. The latter ranked second in the dominance hierarchy established between seven highly invasive ant species, and ranked before *L. humile*. The results observed here when *L. humile* interacted with *La. niger* or *La. neglectus* were therefore unexpected. The fact that the study of Bertelsmeier et al. [21] was based only on dyadic or ten versus ten interactions could explain the differences observed with our results. Although interference competition involves aggressive encounters between workers, including physical and chemical aggressions, it does not imply an important effect at the colony level [46,47]. In our experiments, we used colony fragments constituted by a hundred of workers with brood and queens for both *L. humile* and *La. neglectus*. In these situations, the possibility of many recruits implicated in the fights could thus be the cause of reduced differences in the outcomes of interactions with respect to the results of Bertelsmeier et al. [21]. This is especially relevant for *L. humile*, which often relies on group-level processes to successfully displace other species based on cooperative fighting [7,12,16]. Moreover, irrespective of their position in the dominance hierarchy, *L. humile* can adopt ‘‘the bourgeois strategy’’ during agonistic encounters with other species, changing its behavior based on numerical dominance: lone workers tend to be submissive in encounters, whereas, when numerically dominant, workers are aggressive. *Linepithema humile* has been shown to adopt this strategy against native species such as *P. pallidula, T. group nigerrimum*, or *Monomorium antarcticum* (Smith) [9,18,48]. Moreover, the bourgeois strategy could be a mechanism used by *L. humile* to co-occur with *La. neglectus* [49]. In this sense, because of the recent expansion of *La. neglectus* in Europe, where *L. humile* already invades, it was unexpected that the “resident effect” was not a primary determinant of the behavioral syndrome involved in competitive interactions for these species. Most of the parameters studied suggested that the species’ identity was more important than the status as resident or colonizer. However, we evaluated the resident effect as the access to the foraging arena, without accounting for the advantage of an earlier establishment and the consequent increase in density and numerical dominance [50], which could partly limit our results.

## 5. Conclusions

Our results suggest that native *La. niger* species could impact invasive ant populations of *L. humile* by means of interference competition. Although rare, this report is not the first instance of biotic resistance of native species against Argentine ants through physical aggression or chemicals to successfully defend themselves and their territory (e.g., [39]). Whereas *L. humile* has become invasive in Southern Europe, the invasion process could be slowed down in Northern Europe by such native dominant species rather than by other invasive ones, even under the scenario of co-occurrence between several highly invasive species [21]. However, the antagonistic behavior observed in this study does not necessarily imply that interspecific competition would take place in real conditions, because competition is a process of populations, not of individuals [5]. Further studies on the natural invasion progress and species competitive interactions in the field, especially including the study of long-term interactions, would be needed to eventually extend our conclusions.

## Figures and Tables

**Figure 1 animals-10-02451-f001:**
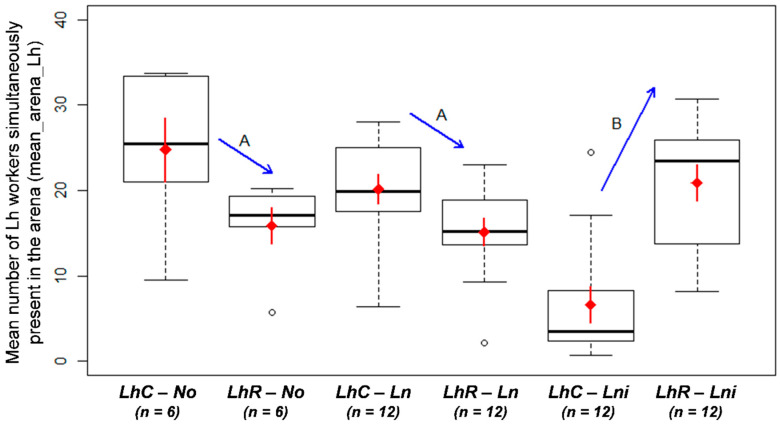
Average number of *Linepithema humile* in the arena during the interaction (mean_arena_Lh; n = 60). LhC: *L. humile* colonizer; LhR: *L. humile* resident; No: no opponent; Ln: *Lasius neglectus* opponent; Lni: *Lasius niger* opponent. Red diamond: mean value; red solid line: standard error of the mean. Letters A, B (blue arrows) indicate significant differences between opponent species × status interaction.

**Figure 2 animals-10-02451-f002:**
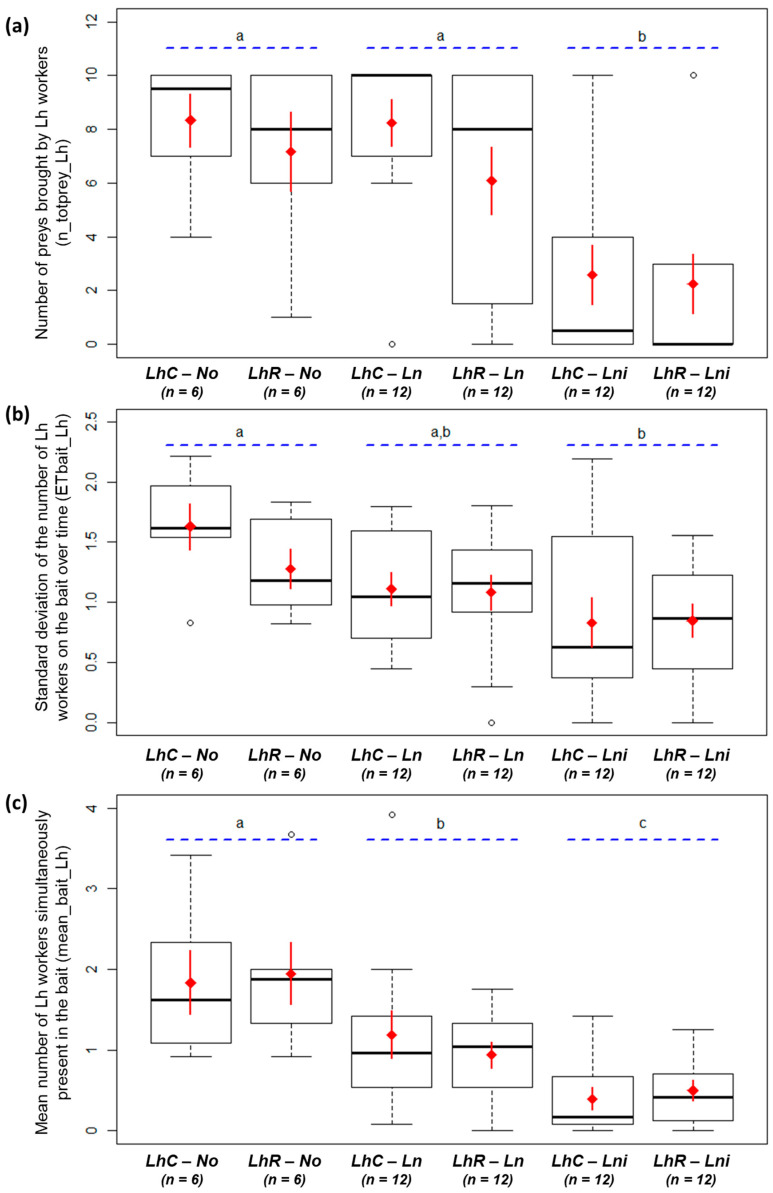
(**a**) Number of preys brought by *Linepithema humile* during the interaction (n_totprey_Lh; n = 60). (**b**) Standard deviation of the numbers of *L. humile* on the bait over time (ETbait _Lh; n = 60). (**c**) Average number of *L. humile* on the bait during the interaction (mean_bait_Lh; n = 60). LhC: *L. humile* colonizer. LhR: *L. humile* resident. No: no opponent; Ln: *Lasius neglectus* opponent, Lni: *Lasius niger* opponent. White dots are outlier individuals; thick black horizontal line: median value; box ends: upper and lower quartiles; whiskers: max and min values. Red diamond: mean value; red solid line: standard error of the mean (SEM). Letters a, b, and c (blue dotted lines) indicate significant differences between opponent species.

**Figure 3 animals-10-02451-f003:**
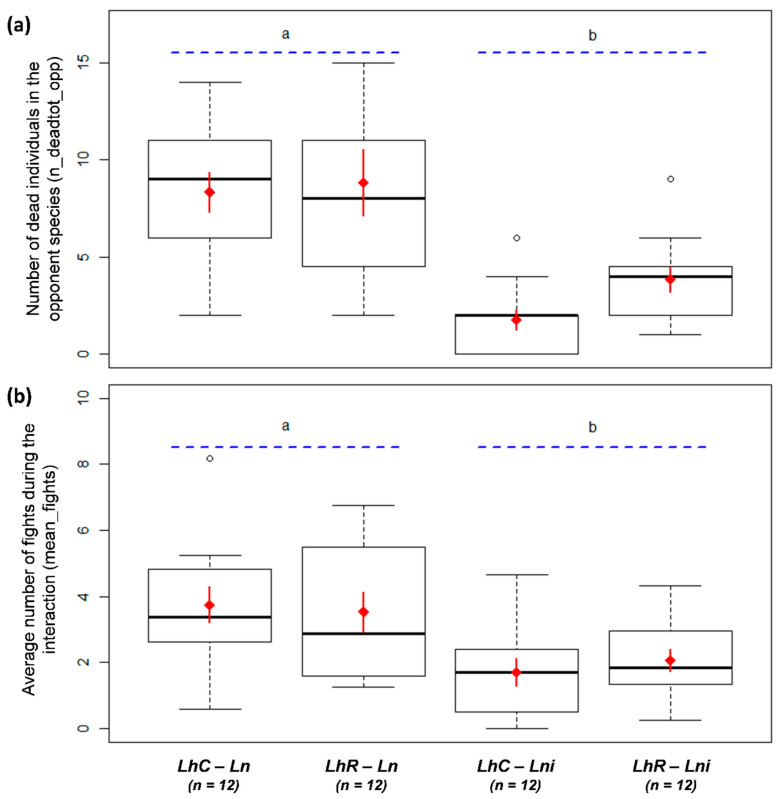
(**a**) Average number of fights during the interaction (mean_fights; n = 48). (**b**) Number of dead individuals in the opponent species (n_deadtot_opp; n = 48). Mean number of dead *Linepithema*
*humile*: 14.08 ± 7.28 when facing *Lasius. neglectus*; 15.08 ± 6.70 when facing *Lasius. niger*. LhC: *L. humile* colonizer. LhR: *L. humile* resident. No: no opponent; Ln: *Lasius neglectus* opponent, Lni: *L niger* opponent. White dots are outlier individuals; thick black horizontal line: median value; box ends: upper and lower quartiles; whiskers: max and min values. Red diamond: mean value; red solid line: standard error of the mean (SEM). Letters a and b (blue dotted lines) indicate significant differences between opponent species.

**Figure 4 animals-10-02451-f004:**
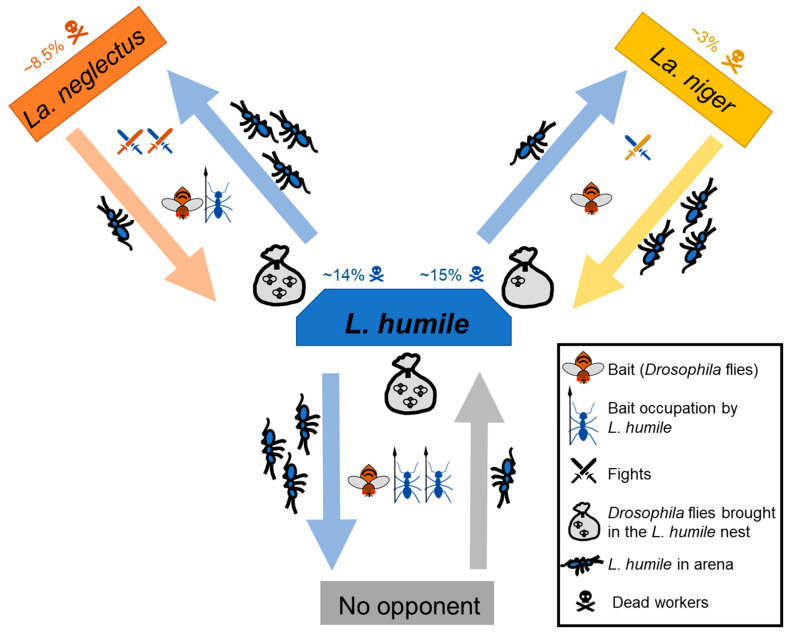
Summary of the patterns observed in the interactions between *Linepithema humile* (blue) and *Lasius neglectus* (orange) or *Lasius niger* (yellow) or without opponent (grey). The arrows point towards the resident species, indicating the direction of the colonization.

**Table 1 animals-10-02451-t001:** Effects of the opponent species (Ln—*Lasius neglectus*, Lni—*Lasius niger* or No—no opponent) and the status of *Linepithema humile* (C—colonizer or R—resident) in the behavioral descriptors. A. Main effects. B. Post hoc tests of the significant variables. Interaction means the interaction between opponent and status. Significant statistics are marked in bold.

A	Opponent		Status		Interaction	
	*χ*2	*p*	*χ*2	*p*	*χ*2	*p*
**M1: Mean_arena_Lh**	6.95	0.03	1.16	0.281	**24.89**	**<0.001**
**M2: n_totprey_Lh**	**32.41**	**<0.001**	1.69	0.194	0.52	0.771
**M3: ETbait_Lh**	**11.44**	**0.003**	0.31	0.578	1.06	0.589
**M4: mean_bait_Lh**	**37.91**	**<0.001**	0.02	0.903	1.13	0.568
M5: n_deadtot_Lh	0.38	0.538	1.73	0.188	0.11	0.738
**M6: n_deadtot_opp**	**30.38**	**<0.001**	2.57	0.109	3.36	0.067
**M7: mean_fights**	**9.21**	**0.002**	0.02	0.887	0.25	0.615
M8: t_maxfights	0.82	0.367	2.46	0.117	2.17	0.141
**B**			*Estimate*	*SE*	*t.ratio*	*p* value
**M1: Mean_arena_Lh**	A: No vs. Ln opponent		5.34	5.73	0.93	0.954
	B: No vs. Lni opponent		13.48	5.69	2.37	0.142
	C: Ln vs. Lni opponent		8.14	4.63	1.76	0.462
	D: C vs. R Status.		−10.72	6.58	−1.63	0.556
	E: Interaction A:D		3.84	5.73	0.67	0.992
	**F: Interaction B:D**		**23.42**	**5.69**	**4.12**	**<0.001**
	**G: Interaction C:D**		**19.58**	**4.63**	**4.23**	**<0.001**
**M2: n_totprey_Lh**	A: No vs. Ln opponent		0.53	0.74	0.71	0.861
	**B: No vs. Lni opponent**		**3.47**	**0.74**	**4.68**	**<0.001**
	**C: Ln vs. Lni opponent**		**2.95**	**0.61**	**4.87**	**<0.001**
**M3: ETbait_Lh**	A: No vs. Ln opponent		0.72	0.37	1.96	0.158
	**B: No vs. Lni opponent**		**1.23**	**0.37**	**3.36**	**0.004**
	C: Ln vs. Lni opponent		0.51	0.3	1.72	0.251
**M4: mean_bait_Lh**	**A: No vs. Ln opponent**		**0.78**	**0.26**	**3.05**	**0.011**
	**B: No vs. Lni opponent**		**1.55**	**0.26**	**6.02**	**<0.001**
	**C: Ln vs. Lni opponent**		**0.76**	**0.21**	**3.64**	**0.002**
**M6: n_deadtot_opp**	**A: Ln vs. Lni opponent**		**2.41**	**0.42**	**5.72**	**<0.001**
	B: C vs. R Status.		−0.86	0.42	−2.04	0.135
	C: Interaction A:B		0.77	0.42	1.83	0.206
**M7: mean_fights**	**A: Ln vs. Lni opponent**		**3.51**	**1.16**	**3.02**	**0.004**

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
