# Peer review of "The Native Ant Lasius niger Can Limit the Access to Resources of the Invasive Argentine Ant"

_animals, 2020, doi:10.3390/ani10122451_

Round 1
Reviewer 1 Report
In this work, the authors aim to examine competition between an invasive ant (Linepithema humile, the Argentine ant) and either a native ant (Lasius niger) or a related, invasive ant (Lasius neglectus). By forming colonies containing an equal number of ants (but not equal mass – the Lasius species are significantly larger than L. humile), they stage colony-fragment competitions over a food resource: a pile of 10 Drosophila. L humile is either put in the resident role (given prior access to the arena) or the colonizer role (no prior access). The authors report that while La. neglectus only slightly reduces Argentine ant foraging success, La. niger is very effective at dominating the food source when it is in the resident role, inflicting heavy losses on the Argentine ant.
This work adds to a growing body of research using dyadic interactions of workers or colonies to examine dominance between ant species. The results are a valuable addition. The sample sizes are good. The language is somewhat rough, and would benefit from some proof-reading, but does not greatly hinder comprehension. The introduction and discussion are adequate. The species choice is interesting and relevant. In all, this work makes a good addition to the literature.
However, the work also suffers from some (perhaps major) limitations, which limit how much ecological validity or relevance they might have. These limitations are to do with the study design.
Firstly, competition was performed as a burst (1 hour) of starved colonies. This poorly reflects reality. Much more interesting would have been long-term competition between colonies (e.g. over a week).
Secondly, the food resource selected was individually-transportable prey items. Items which are light enough to be transported individually are not usually recruited to 1. Thus, the strong recruitment capabilities of L. humile may not have come into play so much. More importantly, the major resources which these ants will dominate, and fight over, are semi-permanent sources of carbohydrates, such as aphid colonies. It is quite possible that L. humile would be more willing to fight over such precious resources than ephemeral prey items. Moreover, scramble competition is much more important for this prey type than contest competition. This may explain some of the surprise results. Contest competition is the dominant force in access to semi-permanent carbohydrate sources.
I would be more comfortable if the title thus read “Native Lasius niger ants can limit the access of invasive Argentine ants to prey resources”. And then, in the abstract, explicitly mention that the competition phase lasted (only) one hour, and likely represented a mix of scramble and contest competition. In the discussion, these issues should be raised as major limitations. Perhaps also a few references to the relative importance of carbohydrate vs protein sources in invasive ant biology.
Finally, I found the statistical analysis very difficult to follow, and the results impenetrable (especially table 1). I admit to not being very familiar with PCAs. Nonetheless, it was not clear to me why PCAs were used to formulate the predictive variables, rather than developing a priori hypotheses and testing these. This is doubly so, since there clearly were a priori hypotheses. PCA, as far as I understand, more appropriate for description and exploration than for deployment in a well-designed experiment with clear manipulations and potential outcomes 2. I also note that the two major principle components together only explained just over half the variation in one case, and under half in the other, implying that the PCA was not fully effective at reducing the dimensionality to a comfortable level. Perhaps two or three simple mixed-effect models, looking at the effect of each treatment on three major proxies for competition (e.g. # flies retrieved, average # workers on feeder, # casualties) would have provided the same insight in a much more straightforward and broadly accessible manner.
Nonetheless, this is a useful contribution, and should be published pending minor revisions. I will end with a series of minor comments.
Minor comments
- Ln 58 – I thought this advantage was considered mostly due to local numerical dominance?
- Ln 68 – were Tapinoma really native ants in this experiment?
- Why was the raw data, code, and full analysis output not provided? A html Rmarkdown output, rscript, and raw data make a very strong contribution to open science, and require very little effort from the authors. I strongly encourage this provision. Indeed, if I could, I would make this a requirement for publication.
- Figure 1 – why not provide pairwise post-hoc comparisons for all groups, and indicate differences with letters? Gives a fuller description, and saves complicated arrows.
- Figure 2 – this figure is so large that the X axis cannot be seen when examining the top panel. Please either split, or provide X axis labels for each panel.
- Figure 4 took a while to understand, but was eventually comprehensible. I advise avoiding the cross on the graves – this does not translate well interculturally. A skull-and-crossbones might work better, or simply a sketch of a dead ant.
- A very relevant study which was not mentioned is 3, which looks at the competitive ability of a different invasive ant (Formica fuscocinerea) against native ants, including Lasius niger. A very similar experimental design was used (with invader vs resident). In this case, L. niger were very quickly expelled from the (sugar) feeder (fig. 2).
References cited
- Robson, S.K., and Traniello, J.F.A. (1998). Resource Assessment, Recruitment Behavior, and Organization of Cooperative Prey Retrieval in the Ant Formica schaufussi (Hymenoptera: Formicidae). J. Insect Behav. 11, 1–22.
- Jolliffe, I.T., and Cadima, J. (2016). Principal component analysis: a review and recent developments. Philos. Trans. R. Soc. Math. Phys. Eng. Sci. 374, 20150202.
- Pohl, A., Ziemen, V., and Witte, V. (2018). Mass Occurrence and Dominant Behavior of the European Ant Species Formica fuscocinerea (Forel). J. Insect Behav. 31, 12–28.
Reviewer 2 Report
The competitive ability of large fragments of workers of Linepithema humile (invasive) against Lasius neglectus (invasive) and Lasius niger (native) were compared using various parameters during laboratory interactions that also accounted for a "resident" effect. It was expected that the invasive Lasius neglectus would outcompete Linepithema humile, since invasive ants are known to both exploit resources and interfere with other species' ability to access resources. However, the native dominant ant Lasius niger was unexpectedly able to decrease the success of the Argentine ant to collect bait and occupy the arena, while the invasive Lasius neglectus was not able to. This study provides an example of a native ant outcompeting an invasive ant, possibly deterring its spread in parts of Europe. This study also shows the importance of using larger colony fragments in competition assays.
I found the study to be interesting and valuable to knowledge on ant interactions, especially for predicting the future spread of invasive species of interest. It was interesting to see a native ant outcompete an invasive ant, and to see differences in interactions of two invasive ants with larger colony fragments.
I think a map of the current distributions of the three species in areas where Lasius niger occurs would be great. At minimum, I suggest including in the text the areas where Linepithema humile and Lasius neglectus have invaded in Europe.
Species names might have to be addressed, especially since the abbreviation "L." is used for both "Linepithema" and "Lasius".
The main correction is lines 275-277, which has a typo that contradicts the main idea of the paper.
Please see attached for specific minor corrections.
Great experiment, data, and paper!
